# Predictive Role of MRI and ^18^F FDG PET Response to Concurrent Chemoradiation in T2b Cervical Cancer on Clinical Outcome: A Retrospective Single Center Study

**DOI:** 10.3390/cancers12030659

**Published:** 2020-03-12

**Authors:** Anna Myriam Perrone, Giulia Dondi, Manuela Coe, Martina Ferioli, Silvi Telo, Andrea Galuppi, Eugenia De Crescenzo, Marco Tesei, Paolo Castellucci, Cristina Nanni, Stefano Fanti, Alessio G. Morganti, Pierandrea De Iaco

**Affiliations:** 1Gynecologic Oncology Unit, Sant’Orsola-Malpighi Hospital, 40138 Bologna, Italy; giulia.dondi@gmail.com (G.D.); eugeniadecrescenzo@gmail.com (E.D.C.); marco.tesei2@gmail.com (M.T.); pierandrea.deiaco@unibo.it (P.D.I.); 2Centro di Studio e Ricerca delle Neoplasie Ginecologiche (CSR) University of Bologna, 40138 Bologna, Italy; andrea.galuppi@aosp.bo.it (A.G.); stefano.fanti@aosp.bo.it (S.F.); alessio.morganti2@unibo.it (A.G.M.); 3Department of Specialized, Diagnostic, and Experimental Medicine, Sant’Orsola-Malpighi Hospital, 40138 Bologna, Italy; manuela.coe@aosp.bo.it; 4Radiotherapy Unit, Sant’Orsola-Malpighi Hospital, 40138 Bologna, Italy; m.ferioli88@gmail.com; 5Nuclear Medicine Unit, Sant’Orsola-Malpighi Hospital, 40138 Bologna, Italy; silvi.telo@gmail.com (S.T.); paolo.castellucci@aosp.bo.it (P.C.); cristina.nanni@aosp.bo.it (C.N.)

**Keywords:** locally advanced cervical cancer, PET/CT, MRI, concurrent chemoradiotherapy, treatment response, follow up

## Abstract

Tumor response in locally advanced cervical cancer (LACC) is generally evaluated with MRI and PET, but this strategy is not supported by the literature. Therefore, we compared the diagnostic performance of these two techniques in the response evaluation to concurrent chemoradiotherapy (CCRT) in LACC. Patients with cervical cancer (CC) stage T2b treated with CCRT and submitted to MRI and PET/CT before and after treatment were enrolled in the study. All clinical, pathological, therapeutic, radiologic and follow-up data were collected and examined. The radiological response was analyzed and compared to the follow-up data. Data of 40 patients with LACC were analyzed. Agreement between MRI and PET/CT in the evaluation response to therapy was observed in 31/40 (77.5%) of cases. The agreement between MRI, PET/CT and follow-up data showed a Cohen kappa coefficient of 0.59 (95% CI = 0.267–0.913) and of 0.84 (95% CI = 0.636–1.00), respectively. Considering the evaluation of primary tumor response, PET/CT was correct in 97.5% of cases, and MRI in 92.5% of cases; no false negative cases were observed. These results suggest the use of PET/CT as a unique diagnostic imaging tool after CCRT, to correctly assess residual and progression disease.

## 1. Introduction

Cervical carcinoma (CC) is the third commonest gynecological cancer in women worldwide [1,2]. In the past, CC was routinely staged with the clinical FIGO system but the new ESGO guidelines introduced and recommended a TNM classification with a FIGO staging, too [3,4,5]. Early CC and locally advanced CC (LACC) represent two different realities with distinct therapeutic approaches and prognosis. Surgery is the preferred approach in the early stage while concurrent chemoradiation (CCRT) is the standard treatment option in LACC [3,6,7,8]. The target of the external beam radiation therapy (EBRT) includes the pelvic lymph nodes, but the irradiation field can be extended to the common iliac and para-aortic region in patients with nodal involvement [9,10]. The residual macroscopic tumor is then boosted with brachytherapy (BRT). CCRT allows local control of the disease in 70%–80% of patients, with 66% and 58% 5-year overall survival (OS) and disease-free survival (DFS) rates, respectively [11,12]. Recurrences occur in 22%–41% of patients, mostly within the first two years after the end of treatment. The recurrence site is more frequently loco-regional while distant metastases are rare [13]. Treatment response is evaluated 3-6 months after the end of CCRT, clinically and by imaging techniques [14]. Currently there is no agreement on the gold standard imaging technique to evaluate a tumor’s response. Data in the literature are lacking and contradictory. MRI and PET/CT represent the most used imaging techniques [15,16] with different purposes: the assessment of response in the primary tumor (T) performed by MRI and assessment of response of the metastases performed by PET/CT [17,18]. In daily practice, usually MRI represents the first imaging method to evaluate response to therapy and PET/CT is mostly performed only if residual disease is suspected by MRI [3]. However, since clear and reliable data in this field are missing and there is no consensus on which technique should be used, the National Comprehensive Cancer Network (NCCN) [10] recommend both MRI and PET/CT as a post-treatment assessment of tumor response in LACC after CCRT [19].

Considering the uncertainty of current evidence, we retrospectively compared the diagnostic performance of ^18^F-FDG PET/CT and MRI in the response evaluation after CCRT in LACC stage T2b.

## 2. Materials and Methods

### 2.1. Population

The clinical data of all patients with CC referred to our Unit of Gynecologic Oncology of Sant’Orsola Hospital of Bologna (Italy) between June 2007 and January 2017 were retrospectively analyzed. Among these we selected patients with stage T2b treated with CCRT.

Inclusion criteria were (a) histologically proven squamous cell cervical cancer and adenocarcinoma performed by biopsy or cone according to the WHO criteria [20]; (b) clinical stage T2b; c) treatment with CCRT; d) a pre-treatment pelvic MRI and total body ^18^F-FDG PET/CT and post-treatment pelvic MRI and total body ^18^F-FDG PET/CT performed in the radiologic service of our institution; and e) adequate follow-up over 24 months.

Exclusion criteria were (a) patients younger than 18 years old; (b) rare (other than squamous or adenocarcinoma) histological type; (c) a previous history of cancer in the last 5 years; (d) other stages different from T2b; (e) previous surgery or chemotherapy; and (f) pre- or post-treatment pelvic MRI and/or total body ^18^F-FDG PET/CT performed in other centers.

All clinical and pathological data were collected and examined, including age, body mass index (BMI), histological type, TNM staging system [5], radiotherapy and chemotherapy administration.

### 2.2. Concurrent Chemo-Radiotherapy Scheme

All patients underwent pelvic 3-dimensional conformal EBRT. In patients with para-aortic lymph node metastases, detected by ^18^F-FDG-PET/CT, the fields were extended up to the level of the renal vessels or even more cranially based on the positive node site. In case of large lymphadenopathy (short axis greater than one centimeter) with high ^18^F-FDG uptake (SUV_max_ > 3), a highly conformed EBRT boost was prescribed.

Cisplatin (40 mg/m^2^) was administered intravenously once a week concurrently with EBRT. After radio-chemotherapy, all patients underwent a BRT boost, using the high dose rate (HDR) or pulsed dose rate (PDR) technique. The EBRT and BRT doses were prescribed according to the International Commission on Radiation Units and Measurements Reports 62 and 38, respectively [21].

### 2.3. Pelvic MRI Image Analysis

A baseline pelvic MRI was performed before the beginning of the treatment. An area with a high signal intensity in the cervix compared to the cervical stroma on the T2-weighted image with enhancement was considered neoplastic tissue in the pelvic MRI. Images were obtained with a high field magnet 1.5 T MRI system (GE, SIGNA LX HD-xt) using phase arrayed body coils. T2-weighted FSE images were obtained in sagittal, axial and oblique, perpendicular to the axis of the uterine cervix, and coronal with a 3–4 mm thickness and interslice gaps of 0.3/0.4 mm (matrix 320 × 224, FOV 28–32, 4NEX, TE 100, TR 3400) planes. LAVA 3D T1-weighted sequences were obtained before and after the intravenous (i.v.) injection of gadolinium contrast medium. Vaginal distension with aqueous gel (60 cc) was used in order to improve evaluation of the vaginal walls. Patients fasted for 4–6 h before the examination in order to reduce bowel motion artefacts and the bladder must be half full.

### 2.4. Total Body ^18^F-FDG PET/CT Image Analysis

Whole body PET and low-dose CT scans were obtained one hour after the i.v. injection of 3.5 MBq/Kg of ^18^F-FDG (PET scanner Discovery PET-CT 710, GE, Boston, Massachusetts, United States). A low dose Computed Tomography (CT) scan was performed both for attenuation correction and to provide an anatomical map. The CT parameters were 120KV, 80mA and 0.8 s for rotation, and a thickness of 3.75 mm. An iterative 3-D ordered subsets expectation maximization method with two iterations and 20 subsets, followed by smoothing (with a 6-mm 3-D gaussian kernel) with CT-based attenuation, scatter and random coincidence event correction, was used to reconstruct the PET images.

Patients fasted for 6 h and eventual insulin therapy was interrupted at least 6 h before the examination. All patients were positioned supine on the imaging table, arms above, and acquired from the base of the skull to the mid thighs. All ^18^F-FDG PET/CT scans were reviewed by two expert nuclear medicine physicians with more than ten years of experience and a specific interest in gynecological malignancies. Discrepancies have been solved by consensus. For each scan, together with a visual assessment, a maximum standardized uptake value (SUV_max_) was measured for every area of the focal uptake higher than the background and suspected to be a metastasis based on qualitative interpretation according to the location, the size and the intensity of the ^18^F-FDG uptake.

### 2.5. Evaluation of the Response to CCRT

All patients were studied at baseline with pelvic MRI and total body ^18^F-FDG PET/CT and treatment response was performed 6 months after the end of CCRT in the same tomographs. All radiological images were reviewed by M.C. for MRI and C.N. and P.C. for PET. Response to treatment was defined as complete response (CR), partial response (PR), progressive disease (PD) and stable disease (SD) according to RECIST criteria for the pelvic MRI and EORTC criteria for the ^18^F-FDG PET/CT [22,23].

### 2.6. Follow Up

Patients defined as CR by both imaging techniques (^18^F-FDG PET/CT and MRI) had a follow-up gynecological examination every 4 months in the first 2 years and every 6 months for 3 years. A chest–abdomen CT scan was performed every 12 months or in case of clinical suspicion of a relapse.

Patients defined not complete responders (PR, SD, PD) by one or both techniques were treated according to the localization of the residual or progressive disease (surgery or chemotherapy) or observed periodically with subsequent clinical or instrumental investigations in case of doubt.

Progression-free survival (PFS) was calculated from the first diagnosis to recurrence and overall survival (OS) was obtained from diagnosis to the last follow up or death.

The study was performed according to Helsinki declaration 2013, and all patients signed an informed consent and the local ethical committee of Sant’Orsola-Malpighi Hospital—Bologna approved this study (CE 322/2019/Oss/AUOBo).

### 2.7. Statistical Analysis

The software IBM SPSS ^®^ 20.0, 2012 (Statistical Package for Social Science), was used for statistical data analysis and a *p*-value < 0.05 was considered statistically significant. Continuous variables were expressed as mean ± SD and categorical variables as percentages. Survival curves were calculated using the Kaplan–Meier method. Cohen’s kappa was used to compare the two imaging techniques.

## 3. Results

### 3.1. Population

The flow chart of the recruitment is showed in Figure 1. In total, 40 patients met the inclusion criteria (patients with histologically proven squamous cell cervical cancer and adenocarcinoma stage T2b treated with CCRT with a pre-treatment pelvic MRI and total body ^18^F-FDG PET/CT and post-treatment pelvic MRI and total body ^18^F-FDG PET/CT performed in the radiologic service of our institution, and with adequate follow-up over 24 months) and were enrolled in the study. The patients’ characteristics are reported in Table 1.

The mean EBRT total dose was 45 ± 1 Gy, eight patients (20%) needed extended-field pelvic and para-aortic radiotherapy and the mean dose of the boost on the positive nodes was 15.1 ± 6.2 Gy. All patients received weekly Cisplatin at the dose of 40 mg/sm. Eight out of 40 patients (20%) received four cycles, 27/40 (67.5%) patients received five cycles and 5/40 (12.5%) patients received six cycles. All patients received a BRT boost with a mean dose of 28.7 ± 6.3 Gy.

### 3.2. MRI Parameters

The mean pre-treatment maximum tumor diameter measured by MRI was 45.5 ± 14.4 mm (mean ± SD). In six patients (15%) MRI detected nodal involvement and the mean node short-axis length was 19.5 ± 8.9 mm (mean ± SD). After treatment, a complete response of the primary cervical tumor was recorded in 32 (80%) patients, while residual disease was observed in eight (20%) patients. Only one metastatic lymph node was still present (2.5%) with a short-axis length of 15 mm.

### 3.3. ^18^F-FDG PET/CT Parameters

Pathological uptake was present in all primary tumors and lymph node involvement was detected in 17/40 patients (43%). The mean pre-treatment SUV_max_ of the primary tumor was 14.2 ± 5.6 (mean ± SD) and the mean SUV_max_ of pathologic lymph nodes was 6.2 ± 2.6 (mean ± SD). After CCRT in 35 target lesions (87.5%) the uptake of *^18^F*-FDG was normalized and the SUV_max_ of the residual lesions was 9.7 ± 5.6 (mean ± SD). After the CCRT persistence, increased SUV_max_ in the lymph nodes was detected in five patients (12.5%), showing a mean SUV_max_ of 7.5 ± 5.1 (mean ± SD).

### 3.4. Agreement between MRI and ^18^F-FDG PET/TC

Agreement between MRI and ^18^F-FDG PET/CT in the evaluation response to therapy was observed in 31/40 (77.5%) of cases (Table 2 and Figure 2A,B). In these cases, a CR was observed in 28/31 patients (90.3%) and follow-up data showed a strong correlation with the response to therapy. In fact, only five out of 28 patients (18%) with CR experienced a subsequent treatment failure: Two local recurrences after 63 months and 64 months, respectively; one lung metastasis after 15 months; one vertebral metastasis after 24 months; and one patient showed lung and cerebellar metastases after 75 months. A PR was seen in 2/31 (6.5%) of the patients, who underwent rescue therapies. One of them underwent radical surgery and the histological exam confirmed residual disease in the cervix and in pelvic lymph nodes. The second one received palliative treatment because of severe comorbidities. Both patients died due to progressive disease after 12 and 18 months, respectively. One patient showed PD (1/31, 3.2%), and underwent a posterior exenteration (radical hysterectomy, bilateral salpingectomy, total colpectomy, pelvic and para-aortic lymphadenectomy, rectal resection with colorectal termino-terminal anastomosis and ileostomy). The patient died after 18 months.

### 3.5. Disagreement between MRI and ^18^F-FDG PET/CT

In 9/40 (22.5%) cases, MRI and ^18^F-FDG PET/CT showed different results in terms of response (Table 2 and Table 3). In Cases #1 and #2, MRI showed a CR, while ^18^F-FDG PET/CT showed a PD due to the detection of distant metastasis in the lung and in a supraclavicular node, respectively. Both patients died after 47 and 33 months.

In Case #3 and #4, MRI showed a PR while ^18^F-FDG PET/CT showed a PD. In particular, in Case #3 the MRI showed a PR while the ^18^F-FDG PET/CT showed a CR of the primary lesion but also a vertebral metastasis. The patient underwent chemotherapy and died after 33 months. In Case #4, both imaging techniques showed a PR at the level of the cervical lesion but ^18^F-FDG PET/CT detected also a common iliac lymph node metastasis. The patient underwent anterior exenteration (radical hysterectomy, bilateral salpingo-oophorectomy, total colpectomy, pelvic and para-aortic lymphadenectomy and total cystectomy with Indiana pouch neobladder reconstruction) followed by chemotherapy. The patient is still alive after 102 months of follow-up with no evidence of disease. The pathological evaluation confirmed the ^18^F-FDG PET/CT findings.

In Cases #5 (Figure 2C,D), #6 and # 7, MRI showed a PR while the ^18^F-FDG PET/CT showed a CR. Patients #5 and #6 are alive after 48 and 67 months of follow-up, respectively, without evidence of disease and patient # 7 died due to unrelated reasons (hearth attack).

In Case #8 both techniques reported CR in the pelvis but a glucose uptake near to the spleen was observed. The patient underwent surgery in order to remove the upper abdominal lesion and the pathological examination showed a desmoids tumor unrelated to CC. The patient is alive after 49 months of follow-up without evidence of disease.

In Case #9, ^18^F-FDG PET/CT showed a borderline increase uptake in the primary tumor (SUV_max_ was 3.2), interpreted as a suspicious persistence of disease (PR), although MRI showed no sign of disease (CR); the patient is alive after 79 months of follow-up without evidence of disease.

### 3.6. Follow-Up Outcomes

According to follow-up data we divided patients in 33 responders (82.5%) and seven non-responders (17.5%) (Table 4). Considering local control and distant metastasis, the number of false positive findings were one for ^18^F-FDG PET/CT (Case #9) and three for pelvic MRI (Cases #5, #6 and #7). The number of false negative findings were two (Cases #1 and #2) for pelvic MRI and none for ^18^F-FDG PET/CT.

The agreement between pelvic MRI, ^18^F-FDG PET/CT and follow-up data showed a Cohen kappa coefficient of 0.59 (95% CI = 0.267–0.913) and of 0.84 (95% CI = 0.636–1.00), respectively.

Considering the evaluation of primary tumor response, ^18^F-FDG PET/CT was correct in 97.5% of cases and MRI in 92.5% of cases; no false negative cases were observed with both methods.

The 5-year PFS and OS rates were, respectively, 78% and 88% (Figure 3 and Figure 4).

## 4. Discussion

Our retrospective study showed a better correspondence with the follow-up data of ^18^F-FDG-PET/CT (0.84) with respect to MRI (0.59) in the evaluation of the response to CCR for primary tumor and distant metastasis in T2b CC.

T2b CC represents 38% of CC with the possibility of salvage surgery and complete response to therapy higher than other LACC stages. The persistence of disease in T2b tumors is important to plan salvage surgery. The surgeon’s aim is to exclude the diagnosis of disease progression rather than macroscopic parametrial invasion since the therapy is not selective on the parametria but is represented by an exenteration surgery. In the literature, the role of MRI and ^18^F-FDG-PET/CT before treatment in CC seems well defined. MRI is superior for the assessment of parametrial, vaginal, cervical, bladder and rectum involvement [24,25]. ^18^F-FDG-PET/CT, instead, is more sensitive in detecting lymph node metastases (pelvic, paraaortic, inguinal and supraclavicular) and peritoneum, mesentery, gastrointestinal tract, pleura and mediastinal involvement [26]. Therefore, both exams are useful in the treatment choice (surgery or CCRT), and in RT planning [27,28]. However, evidence on the evaluation of response to CCRT are weak, although this assessment has a crucial role to decide subsequent treatments [29,30,31,32]. To detect residual disease, both clinical and radiological competences are needed. Unfortunately, gynecologic examination after RT is difficult to perform. Indeed, vaginal adhesions and post-RT fibrosis interfere with an accurate visualization of the cervix and with a thorough evaluation of the parametria [33]. Post-RT MRI scans may not optimally evaluate treatment response due to edema/inflammation in T2W hyperintense areas, as well as heterogeneous Gd-contrast enhancement from the main lesion [34]. On the other hand, post radiotherapy necrosis and inflammation may interfere with ^18^F-FDG-PET/CT evaluation [35]. Therefore, well-defined roles of the two imaging techniques after CCRT are lacking.

In the assessment of primary tumor response, our data found that ^18^F-FDG-PET/CT is more sensitive (100% versus 80%) and specific (97% versus 89%) than MRI. In fact, three cases correctly diagnosed by PET as CR were false positive using MRI (#5, #6 and #7). When we analyzed the false positive cases found with MRI, we found that probably they were due to the residual fibrosis subsequent to RT as reported in literature [36]. Based on these data, if we had performed only MRI, 20% of our patients would have received an incorrect diagnosis, while with ^18^F-FDG-PET/CT this percentage would have been of only 3%. False positive results are an important issue: it could require subsequent exams, biopsies and the possibility of over-treatments and complications. These data are supported by a study concluding that MRI accuracy is not enough to select patients who can benefit from completion surgery if residual disease is suspected due to the high false positive rate [37]. In our experience, residual disease with MRI or ^18^F-FDG-PET/CT after CCRT must be carefully considered before proceeding with further invasive investigations. Instead, in case of negative imaging, no doubt must arise because no false negative cases were observed with both techniques.

The incidence of metastatic progression of disease in LACC is not negligible, particularly in the upper abdomen and chest, even in patients with primary tumor controlled by CCRT. Indeed, in our series, in 10% of cases ^18^F-FDG PET/CT showed distant metastases (PD) not detectable by pelvic MRI. In our experience, sensitivity and specificity of ^18^F-FDG PET/CT when we considered distant metastasis was 100% and 97%, respectively; these values were higher than for MRI (71% and 91%, respectively). These values are similar to data from studies reporting 82%–100% and 78%–100% sensitivity and specificity rates for MRI, respectively, and 83%–100% and 50%–100% for ^18^F-FDG PET/CT, respectively [38,39]. The sensitivity and specificity of PET increases for distant metastases (86% and 100%, respectively). These data highlighted the high diagnostic value of ^18^F-FDG PET/CT and the importance of a total body evaluation in order to exclude progressive disease even if local control is achieved.

Based on this retrospective analysis we suggest that, after CCRT, ^18^F-FDG PET/CT is effective in response evaluation and that MRI should only be reserved for special cases. It is not easy to compare the results of our study with those of the literature. To our knowledge there is a lack of studies that focused on the direct comparison between 18F-FDG PET/CT and MRI in in the response evaluation after CCRT in LACC stage T2b.

Waldestrom et al. examined 25 LACC patients (stages IB2–IIIB) with both MRI and ^18^F-FDG PET/CT before and after CCRT and found that in almost half of the patients the ^18^F-FDG-PET/CT before treatment provided additional diagnostic information leading to changes in treatment planning compared to information from MRI. MRI, instead, detected pelvic tumor spread not seen on the ^18^F-FDG-PET/CT in 2/24 patients. [40]. On the contrary, a metanalysis of 15 studies of Maeds et al. evaluated the diagnostic accuracy of additional whole body ^18^F-FDG PET/CT compared with conventional imaging in women with suspected recurrent/persistent cervical cancer, concluding that the use of ^18^F-FDG PET/CT in recurrent cervical cancer and its endorsement by national guidelines is not supported by the literature. However, none of the studies included in this metanalysis directly compared additional PET–CT with MRI or CT separately, and the included populations were very heterogeneous and often follow-up periods were short. [41].

The results of other studies are generally based on the analysis of a single imaging technique, ^18^F-FDG PET/CT [42,43] or CT/MRI [44], or reported that the addition of ^18^F-FDG PET/CT to other imaging techniques increases the sensitivity and specificity [45]. Finally, no studies compared ^18^F-FDG PET/CT with MRI and all studies had a short follow-up period. This fact represents an evident problem considering that follow-up accuracy and duration are important to confirm the radiological results, being a pathological response not available in case of CCRT. On the contrary, our study is supported by long-term follow up of 50 ± 26 months (mean ± SD), with overall survival results (Figure 3) consistent with data in the literature.

Some considerations about false positive of ^18^F-FDG PET/CT are required. Case #8 developed a new and benign tumor, not related to the LACC. We considered this case as FP but probably it should be better considered as a concomitant disease. Case #9 was a borderline glucose case, which was not considered as a positive lesion and therefore the patient received no subsequent treatments.

Limitations of our study were the retrospective nature of the analysis and the small sample size, while strengths of the study could be considered the prolonged follow-up and the well-defined setting of patients. Further studies are needed to confirm our findings in a prospective setting.

## 5. Conclusions

In conclusion, ^18^F-FDG PET/CT seems to be superior to pelvic MRI in the evaluation of the treatment response to CCRT in T2b CC. In addition, in some cases ^18^F-FDG PET/CT detected distant metastasis resulting in a change in the therapeutic strategy. Further studies are needed to confirm that ^18^F-FDG PET/CT should be the standard option, almost six months after the end of CCRT in LACC, to evaluate the treatment response. These results may suggest that ^18^F-FDG-PET/CT is a unique diagnostic imaging tool to use after CCRT in order to correctly assess residual and progression disease.

## Figures and Tables

**Figure 1 cancers-12-00659-f001:**
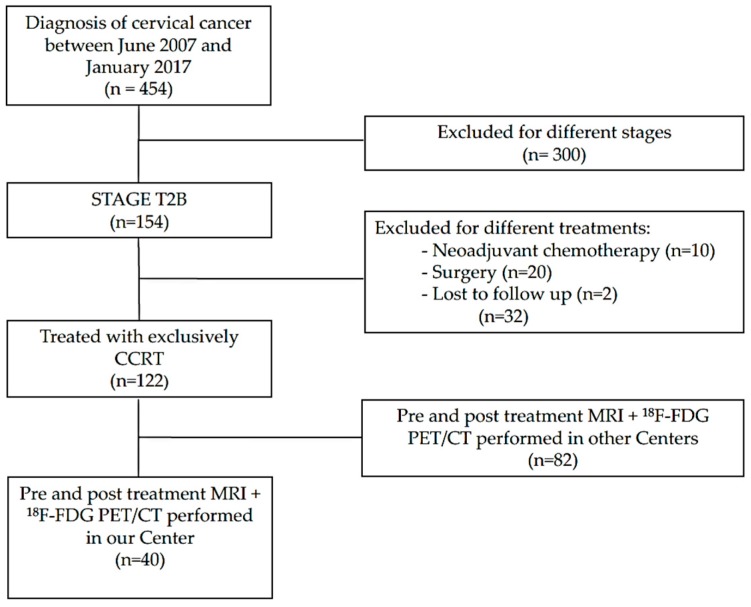
Flow chart of the study. Patient selection from our database of patients with cervical cancer.

**Figure 2 cancers-12-00659-f002:**
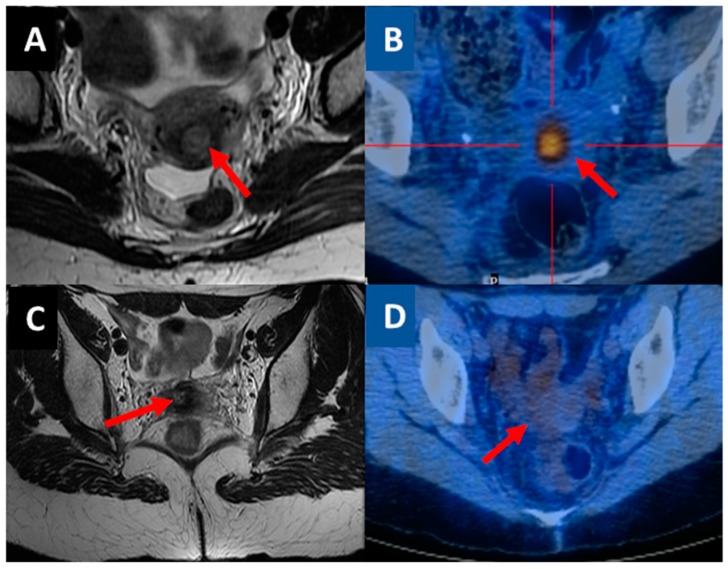
(**A**) and (**B)** show agreement between MRI and ^18^F-FDG PET/CT after CCRT. The red arrows indicate the residual tumor (partial response). (**C**) and (**D**) show disagreement between MRI and ^18^F-FDG PET/CT after CCRT. In panel (**C**) the red arrow indicates the residual tumor (partial response) and in (**D**) the red arrow indicates the absence of a tumor (complete response) in MRI and^18^F-FDG PET/CT, respectively.

**Figure 3 cancers-12-00659-f003:**
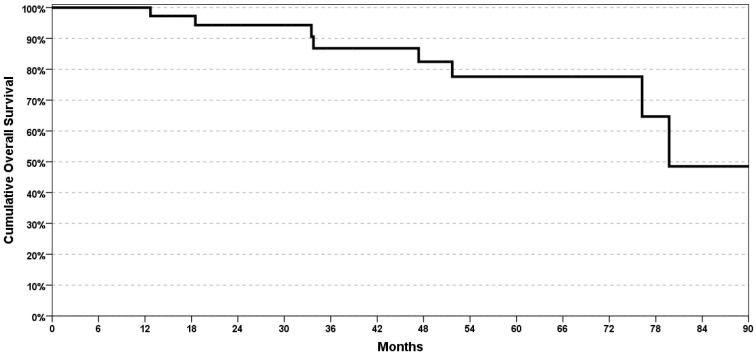
Kaplan–Mayer analysis of overall survival (OS).

**Figure 4 cancers-12-00659-f004:**
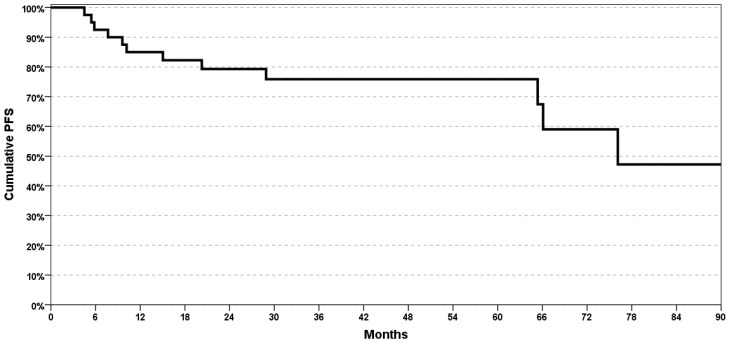
Kaplan–Mayer analysis of progression-free survival (PFS).

**Table 1 cancers-12-00659-t001:** The population’s characteristics.

Characteristics	Values
*N*	40
Age, years (mean ± SD)	61 ± 16
Mean age at diagnosis (mean ± SD)	55 ± 15
BMI mean (mean ± SD)	24.5 ± 4.2
*Histotype*	
Squamous *n* (%)	32 (80%)
Adenocarcinoma *n* (%)	8 (20%)
*Stage TNM*	
*T2b*	40 (100%)
*N0*	23 (57%)
*N1*	17 (43%)
*M 0*	40 (100%)

Note: *n*: number of patients; BMI: body mass index; T: primary tumor; N: nodes; M: metastases.

**Table 2 cancers-12-00659-t002:** Concordance between MRI and FDG-PET/CT after CCRT.

MRI	^18^F-FDG PET/CT				
	CR	PR	SD	PD	Total
CR	28	2	0	2	32
PR	3	2	0	2	7
SD	0	0	0	0	0
PD	0	0	0	1	1
Total	31	4	0	5	40

Note: CCRT: concomitant chemo-radiotherapy; CR: complete response; PR: partial response; SD: stable disease; PD: progression disease.

**Table 3 cancers-12-00659-t003:** Details of discordant cases.

Case	^18^F-FDG PET/CT	MRI	FU	Localization of Disease and Treatment	Status to the Last FU
1	PD	CR	PD	Lung metastasis (PET) → chemotherapy	DOD
2	PD	CR	PD	Supraclavicular node metastasis (PET) → chemotherapy	DOD
3	PD	PR	PD	Vertebral metastasis (PET) → chemotherapy	DOD
4	PD	PR	PD	Common iliac lymph node metastasis (PET) → anterior pelvectomy and chemotherapy	NED
5	CR	PR	CR	Follow up → no disease	NED
6	CR	PR	CR	Follow up → no disease	NED
7	CR	PR	CR	Follow up → Died for hearth attack	DOC
8	PR	CR	CR	Uptake near the spleen→ Abdominal surgery: desmoid tumor. Follow up → no disease	NED
9	PR	CR	CR	Follow up → no disease	NED

Note: CCRT = concomitant chemo-radiotherapy; CR = complete response; PR = partial response; SD = stable disease; PD = progressive disease; DOD = Died of disease; DOC = Died of other causes; NED = no evidence of disease; FU = follow up.

**Table 4 cancers-12-00659-t004:** Accuracy in the treatment response evaluation of MRI and FDG-PET/CT compared to the follow-up data.

Response	FU data	^18^F-FDG PET/CT	MRI
Responders (CR)	33 (82.5%)	31 (77.5%)	32 (80%)
Non-responders (PR + SD + PD)	7 (17.5%)	9 (22.5%)	8 (20%)
False Positive Findings	-	1	3
False Negative Findings	-	0	2

Note: CCRT: concomitant chemo-radiation therapy; CR: complete response; PR: partial response; SD: stable disease; PD: progressive disease; FU: follow up.

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
