# Peer review of "Predictive Role of MRI and 18F FDG PET Response to Concurrent Chemoradiation in T2b Cervical Cancer on Clinical Outcome: A Retrospective Single Center Study"

_cancers, 2020, doi:10.3390/cancers12030659_

Round 1

Reviewer 1 Report

The authors present a comprehensive evaluation between MRI and PET/CT, demonstrating the utility of the later as a precise diasgnostic tool for cervical cancer.

However, could the authors at least show one pair of images from the same patient and highlight the lesion regions for both modality?

Author Response

Dear Reviewer, thank you for your comment and thank you for your precious suggestions.

Point 1) Could the authors at least show one pair of images from the same patient and highlight the lesion regions for both modality?

Response 1: Figure 2 was added to the manuscript; in the figure 2A and 2B we showed a case with agreement between RMN and PET after CCRT, and in the figure 2C and 2D we showed a case of disagreement between the two methods.

Reviewer 2 Report

Dear Editor,

the paper of Anna Myriam Perrone et al. reported interesting data about the predictive role of MRI and 18F FDG PET response to 2 concurrent chemoradiation in T2b cervical cancer.

I  my opinion this study is innovative and can improve the current management of cervical cancer patient.

Therefore, I suggest the publication of the paper after the following minor revision

Minor Revision

Add more specifications about the methodology of both PET and MRI scan

Specify in detail which Helsinki declaration is referred to.

The Flow chart image is fade

Were there cases of discordance between nuclear medicine ? if yes, how have they been resolved?

Some panel with histological, MRI and PET images can improve the manuscript

Author Response

Dear Reviewer, thank you very much for your comment

As you required, we added:

Point 1) Add more specifications about the methodology of both PET and MRI scan

Response to point 1: In materials and methods paragraph 2.3 about MRI we added “compared to the cervical stroma” in line 94 and “high field magnet” in line 95 and “using phase arrayed body coils” in line 96 and “and the bladder must be half full” in line 102. 

In materials and methods paragraph 2.4 about PET we added:” A low dose Computed Tomography (CT) scan was performed both for attenuation correction and to provide an anatomical map, CT parameters were 120KV, 80mA, 0.8 sec for rotation and thickness of 3.75 mm. An iterative 3-D ordered subsets expectation maximization method with two iterations and 20 subsets, followed by smoothing (with a 6-mm 3-D gaussian kernel) with CT-based attenuation, scatter and random coincidence event correction was used to reconstruct PET images” from line 106 to line 110 and “All patients were positioned supine on the imaging table, arms above, and acquired from the base of the skull to the mid thighs” in lines 112-113.

Point 2) Specify in detail which Helsinki declaration is referred to.

Response to point 2: We referred to Helsinki declaration 2013 and this was added in the text line 135.

Point 3 The Flow chart image is fade

Response to point 3: The flow chart has been modified (line 154)

Point 4: Were there cases of discordance between nuclear medicine? if yes, how have they been resolved?

Response to point 4: Discrepancies have been solved by consensus. And this statement was added in the text in line 115

Point 5: Some panel with histological, MRI and PET images can improve the manuscript

Response to point 5: Figure 2 was added to the manuscript, in the figure 2A and 2B we showed a case of agreement between RMN and PET after CCRT, and in the figure 2C and 2D we showed a case of disagreement between the two methods. Histological specimen was not available in these cases.